# Selective Photo-Assisted Eradication of Triple-Negative Breast Cancer Cells through Aptamer Decoration of Doped Conjugated Polymer Nanoparticles

**DOI:** 10.3390/pharmaceutics14030626

**Published:** 2022-03-12

**Authors:** Luis Exequiel Ibarra, Simona Camorani, Lisa Agnello, Emilia Pedone, Luciano Pirone, Carlos Alberto Chesta, Rodrigo Emiliano Palacios, Monica Fedele, Laura Cerchia

**Affiliations:** 1Instituto de Biotecnología Ambiental y Salud (INBIAS), Universidad Nacional de Río Cuarto y CONICET, Río Cuarto X5800BIA, Argentina; 2Departamento de Biología Molecular, Facultad de Ciencias Exactas, Fisicoquímicas y Naturales, Universidad Nacional de Río Cuarto, Río Cuarto X5800BIA, Argentina; 3Institute of Experimental Endocrinology and Oncology “G. Salvatore” (IEOS), National Research Council (CNR), 80131 Naples, Italy; s.camorani@ieos.cnr.it (S.C.); lisa.agnello@ieos.cnr.it (L.A.); mfedele@unina.it (M.F.); 4Institute of Biostructures and Bioimaging, National Research Council (CNR), 80145 Naples, Italy; emiliamaria.pedone@cnr.it (E.P.); luciano.pirone@cnr.it (L.P.); 5Instituto de Investigaciones en Tecnologías Energéticas y Materiales Avanzados (IITEMA), Universidad Nacional de Rio Cuarto y CONICET, Río Cuarto X5800BIA, Argentina; cchesta@exa.unrc.edu.ar (C.A.C.); rpalacios@exa.unrc.edu.ar (R.E.P.); 6Departamento de Química, Facultad de Ciencias Exactas, Fisicoquímicas y Naturales, Universidad Nacional de Río Cuarto, Río Cuarto X5800BIA, Argentina

**Keywords:** conjugated polymer nanoparticles, aptamer, photodynamic therapy, TNBC, cancer targeting, chemotherapy resistance

## Abstract

Photodynamic therapy (PDT) may be an excellent alternative in the treatment of breast cancer, mainly for the most aggressive type with limited targeted therapies such as triple-negative breast cancer (TNBC). We recently generated conjugated polymer nanoparticles (CPNs) as efficient photosensitizers for the photo-eradication of different cancer cells. With the aim of improving the selectivity of PDT with CPNs, the nanoparticle surface conjugation with unique 2’-Fluoropyrimidines-RNA-aptamers that act as effective recognition elements for functional surface signatures of TNBC cells was proposed and designed. A coupling reaction with carbodiimide was used to covalently bind NH_2_-modified aptamers with CPNs synthetized with two polystyrene-based polymer donors of COOH groups for the amide reaction. The selectivity of recognition for TNBC membrane receptors and PDT efficacy were assayed in TNBC cells and compared with non-TNBC cells by flow cytometry and cell viability assays. Furthermore, in vitro PDT efficacy was assayed in different TNBC cells with significant improvement results using CL4, sTN29 and sTN58 aptamers compared to unconjugated CPNs and SCR non-specific aptamer. In a chemoresistance TNBC cell model, sTN58 was the candidate for improving labelling and PDT efficacy with CPNs. We proposed sTN58, sTN29 and CL4 aptamers as valuable tools for selective TNBC targeting, cell internalization and therapeutic improvements for CPNs in PDT protocols.

## 1. Introduction

According to the International Agency for Research on Cancer, an estimated 19.3 million new cancer cases and almost 10 million cancer-related deaths occurred in 2020 [1]. Breast cancer is the leading cause of cancer-related deaths among women globally and has surpassed lung cancer as the most commonly diagnosed cancer, with an estimated 2.3 million new cases (11.7%) [1]. Among breast cancer subtypes, triple-negative breast cancer (TNBC) is the deadliest form because it is more aggressive, usually diagnosed at a later stage and more likely to develop recurrence after conventional treatments [2]. TNBC accounts for 10–15% of all breast cancer cases and, due to the lack of estrogen/progesterone receptors (ER/PR) and the low-level expression of human epidermal growth factor receptor 2 (HER2), does not respond to hormonal or anti-HER2 therapies [3]. Currently, chemotherapy remains the mainstay of treatment for TNBC patients; however, it involves issues such as high toxicity and high failure rate due to the induced non-specific distribution of drugs and rapidly acquired drug resistance [4,5]. Thus, there is a need for new therapeutic compounds or treatments to eliminate, in a more selective way, TNBC tumor cells.

Given the intrinsic spatial selectivity of photo-assisted therapies, and their compatibility with other therapeutic options, photodynamic therapy (PDT) has been used as either the main therapy or an adjuvant therapy for the treatment of many solid tumors, including breast cancer [6,7,8]. PDT requires three components: a photosensitizer (PS), which localizes in the tumor tissue, a light source of the appropriate wavelength to photo-excite the PS accumulated in the tumor tissue and dissolved molecular oxygen. With the combination of these elements into tumor cells, it is possible to photogenerate reactive oxygen species (ROS) in a temporally and spatially controlled fashion. These species disrupt the normal redox status of living cells producing lethal cellular damage [9]. PDT has been examined experimentally to be used in breast cancer treatment, including TNBC management, as a main treatment or in combination with other therapeutic approaches [10,11,12]. PDT for TNBC treatment offers reduced long-term mobility, very limited side effects, and better specificity over surgery, chemotherapy or radiotherapy. In addition, if the PS displays bright fluorescence emission after illumination, photodynamic diagnosis (PDD) could be useful to delimitate tumoral tissue before surgery, and also to identify remaining tumor cells after the same procedure [13,14]. On the other hand, it has been reported that PDT can trigger different mechanisms to eliminate tumors [7], and more specifically in breast cancer, PDT has been proposed as a therapeutic option in which the photodynamic effect plays a role in bypassing and inhibiting escape pathways in multidrug resistant cells [15]. For these reasons and also due to the recent development of simpler and more effective irradiation apparatus and of multifunctional and selective nanoparticulated third-generation PSs, there is currently increased interest in the use of PDT protocols for breast cancer treatment.

A considerable number of third-generation PSs have been developed. Among them, conjugated polymer nanoparticles (CPNs) have been demonstrated to act as excellent PSs in generating cytotoxic ROS singlet oxygen (^1^O_2_). These nanoparticulated PSs have advantages compared to molecular PSs, such as extinction coefficients several orders of magnitude higher than molecular PSs, and can be easily modified by attaching bioactive molecules to their surface for targeting. The doping of CPNs with molecular PSs has originated a new type of donor-acceptor nanoparticulated PS to photogenerate ^1^O_2_ more efficiently than either the neat CP or the small molecular dopant components [16,17]. These improvements in the synthesis of CPNs have allowed for their efficient PDT application in various tumor types, such as brain, colorectal, hepatocarcinoma, lung, etc. [18,19,20]. Additionally, CPN biocompatibility in vitro and in vivo has been reported, which encouraged us to continue evaluating this type of nanomaterial for clinical use [19,21,22]. In accordance with the development of CPNs that specifically target cancer cells, thus reducing adverse side effects while improving therapeutic efficacy, different approaches have been considered to conjugate these types of nanoparticles to highly selective recognition molecules, such as antibodies or peptides, against cell-membrane receptors overexpressed on cancer cells [23,24,25]. The exquisite selectivity of oligonucleotide aptamers for cancer cell targeting and their ability to actively internalize into target cells via receptor-mediated endocytosis [26] make these biomolecules excellent candidates to improve the cancer cell labelling capacity and PDT efficacy. We recently used the anti-EGFR CL4 aptamer [27] to confer tumor-targeting properties to cisplatin-loaded polymeric nanoparticles, which were used to treat mice bearing TNBC xenografts, thus overcoming the poor bioavailability of the drug [27]. Furthermore, a group of anti-TNBC nuclease-resistant RNA aptamers were generated by cell-SELEX and were shown to bind with high affinity and specificity to cell-surface receptors unique to TNBC cells [28]. These aptamers have been recently optimized by reducing their size to minimal variants that still preserve the efficacious targeting, rapid cell uptake and anti-tumor properties of the parental moieties, thus representing good candidates to enable active targeting [29]. In this work, we develop optimized CPNs covalently conjugated with either CL4 or two different TNBC aptamers (sTN58, sTN29) for specific labelling and selectively elimination of TNBC cells. We evaluated different nanoformulations to optimize the conjugation of CPNs with aptamers employing amphiphilic polymers (having -COOH groups) and modified aptamers (having NH_2_ groups) to achieve a strong covalent binding. The resulting aptamer–CPNs conjugates were evaluated for the selective labelling of TNBC cells, cell internalization, biocompatibility and in vitro PDT efficacy. Moreover, we demonstrated the ability of PDT protocols using aptamer-decorated CPNs to selectively kill cisplatin-resistant TNBC cells as an attractive treatment alternative to conventional chemotherapy.

## 2. Materials and Methods

### 2.1. Materials

The fluorescent CP poly(9,9-dioctylfluorene-alt-benzothiadiazole) (F8BT, Mn = 70,000 g/mol, PDI = 2.4, ADS Inc., Saint-Sauveur, QC, Canada), the comb-like polymer, polystyrene grafted with ethylene oxide functionalized with carboxyl groups (PS-PEG-COOH, backbone Mn = 6500 g/mol, branches of Mn = 4600 g/mol, Polymer Source Inc, Dorval, QC, Canada), the amphiphilic functional polymer poly(styrene-co-maleic anhydride) (PSMA, terminated by cumene, content of 68% styrene, average molecular weight about 1700 g/mol, Sigma Aldrich, St. Louis, MO, USA) and the porphyrin Pt(II) octaethylporphyrin (PtOEP, >95%, Frontier Scientific, Logan, UT, USA) were used for nanoparticle preparation as previously described by our group [18,19]. Tetrahydrofuran (THF, HPLC grade, Cicarelli, Santa Fe, Argentina) was refluxed for 5 h with potassium hydroxide pellets (KOH, pro-analysis grade, Taurus) and subsequently distilled. Ultrapure water was obtained using Milli-Q^®^ Reference Water Purification System (Merck Millipore, Burlington, MA, USA).

NH_2_-terminated 2’-Fluoropyrimidines (2’F-Pys)-containing RNA CL4, sTN58, sTN29 and scrambled (SCR) aptamers were synthesized by LGC Biosearch Technologies (Risskov-Denmark).
CL4:5′GCCUUAGUAACGUGCUUUGAUGUCG AUUCGACAGGAGGC3′.sTN58: 5′GGACAUAUGAUGCAACGUUGUGGUCCCGUUUGCACUUUGUUUACG3′.sTN29: 5′GGAAGAGAAGGACAUAUGAUCCUGCCCCAACCAUCGCUUCC3′.SCR: 5′UUCGUACC GGGUAGGUUGGCUUGCACAUAGAACGUGUCA3′.

### 2.2. Synthesis of Particles

For binding experiments, F8BT (500 mg/L) and PS-PEG-COOH (2000 mg/L) or PSMA (2000 mg/L) were dissolved in distilled THF. The solutions were mixed to a final concentration of 50 mg/L of F8BT and 10 mg/L of PS-PEG-COOH or PSMA. CPNs stabilized with PS-PEG-COOH or PSMA in aqueous solution were prepared by the nanoprecipitation method [18,19], and particles were labeled as CPN-PSPEG for CPN formed with 16.7% PS-PEG-COOH and CPN-PSMA for CPN formed with 16.7% PSMA. Briefly, 5 mL of the F8BT/PS-PEG-COOH or PSMA solution in THF was quickly added to 10 mL of H_2_O while sonicating (PS-30A, Arcano, Buenos Aires, Argentina). Later, THF and H_2_O were removed under reduced pressure (using a rotary evaporator) yielding a final volume of 5 mL. Lastly, the concentration of F8BT in the resulting dispersion was recalculated by comparing absorption spectra before and after injection in H_2_O. Unless otherwise noted, given particle concentrations are expressed in terms of F8BT mass concentration.

For PDT experiments, porphyrin-doped CPNs stabilized with PS-PEG-COOH or PSMA were prepared, incorporating PtOEP in the pre-injection THF solution to a final concentration of 50, 10 and 5 mg/L of F8BT, PSMA or PS-PEG-COOH and PtOEP [18,30].

### 2.3. Aptamer Conjugation

To bind aptamers to the CPN surface, an EDC-catalyzed reaction was chosen based on previous reports of efficient conjugation of different CPNs with biomolecules [25,31]. EDC reacts with carboxylic acid groups from polymers PS-PEG-COOH or PSMA to form an active *O*-acylisourea intermediate that is easily displaced by nucleophilic attack from primary amino groups, such as those presented in modified NH_2_-aptamers. Briefly, 1 mL of CPN solution (50 mg/L) was mixed with 20 μL HEPES buffer solution (1 M, pH 7.2), different concentrations of activated aptamer solution (100 and 800 pmoL), and 40 μL of EDC solution (5 mg/mL), and the above mixture was left on a rotary shaker overnight at room temperature (RT). After that, aptamer–CPN solutions were concentrated using a centrifugal ultrafiltration tube (Vivaspin Turbo 15, 50,000 MWCO PES, Sartoriuos, Gottingen, Germany) to eliminate unconjugated aptamers, and then washed with MilliQ water and concentrated again to obtain a final solution of 50 mg/L CPN devoid of non-bound (free) aptamers. Finally, aptamer–CPN solution was filtered using a syringe filter (nylon membrane, 13 mm, 0.1 μm pore size, Nazionale, Italy) to eliminate aggregates and sterilize the nanoparticle solution and stored at 4 °C until use. Nanoparticles having a 1:0.17:0.1 F8BT:PSMA:PtOEP mass ratio and decorated with the different TNBC-specific aptamers or SCR were chosen for further evaluation in PDT protocols. Lastly, the concentration of F8BT in the resulting dispersion was recalculated by comparing absorption spectra.

### 2.4. Aptamer-Decorated CPN Characterization

Aptamer-decorated CPNs and nonconjugated CPNs were characterized by size and zeta potential by dynamic light scattering (DLS) using Malvern Zetasizer Nano ZS instrument (Herrenberg, Germany) equipped with a 633 nm laser. UV-VIS absorption spectra were recorded on a diode-array spectrophotometer (Agilent Hewlett-Packard, HP 8452A, Agilent Technologies Inc., Santa Clara, CA, USA) in 1 cm quartz cuvettes at RT. Emission measurements were acquired from dilute solutions (Abs max < 0.1) in 1 cm path length cuvettes at RT and with excitation at the sample absorption maximum. Corrected emission spectra were recorded with a research spectrofluorometer (Fluoromax-4, Horiba, Kyoto, Japan). Agarose gel electrophoresis was used to verify the successful conjugation of CPNs with aptamers.

### 2.5. Indirect Detection of ROS in Aptamer-Decorated CPN Solutions

Detection of singlet oxygen (^1^O_2_) in solution was assayed using the probe 3-[10-(2-carboxyethyl)anthracen-9-yl]propanoic acid (ADPA), a well-known chemical ^1^O_2_ trap [16]. It is well known that ADPA reacts efficiently with ^1^O_2_ to form an endoperoxide (ADPA-O_2_). To this purpose, ADPA was dissolved in a mixture of the different CPNs with a concentration of 12 mg/L and then irradiated for 5 min using a blue LED (λ_ex_~467 nm, FWHM~28 nm, optical power~24 mW) while simultaneously monitoring the change in the absorbance of ADPA as function of time. The oxidation of ADPA was conveniently followed by monitoring changes in absorption at 400 nm directly assigning these changes to variations in ADPA concentration [16]. Each absorption value was first corrected by subtracting the initial absorption of the CPN sample at the same wavelength, and the resulting values were later normalized to the absorption at t = 0 s (Abs_0_) and plotted as Abs/Abs_0_ vs. t. Control experiments under otherwise identical conditions show that: absorption of CPN in the absence of ADPA remains constant and absorption of ADPA in the absence of CPN remains constant.

### 2.6. Cell Lines and Culture Conditions

Human TNBC: MDA-MB-231 and BT-549 cells and triple-positive breast cancer (TPBC, ER^+^, PR^+^, HER2 over-expression) BT-474 (cell line used for the counterselections of the evaluated aptamers in Cell-SELEX procedure) were purchased from the American Type Culture Collection (ATCC, Manassas, VA, USA). MDA-MB-231 cisplatin-resistant (MDA-MB231/cis) cells were generated by treating cells with cisplatin chronically as previously described [28]. MDA-MB-231, BT-549 and MDA-MB231/cis were grown in Roswell Park Memorial Institute-1640 medium (RPMI-1640, Sigma-Aldrich, St. Louis, MO, USA) supplemented with 10% fetal bovine serum (FBS, Sigma-Aldrich). BT-474 were grown in HybriCare medium (ATCC, Manassas, VA, USA) supplemented with 10% FBS. All cells were maintained in a 5% CO_2_ atmosphere at 37 °C.

### 2.7. In Vitro Uptake Analysis by Flow Cytometry

Aptamer-decorated CPN uptake was determined by flow cytometry as previously described [19]. To this end, MDA-MB-231, BT-549, BT-474 and MDA-MB-231/cis were seeded in a 24-well plate (50,000 cell/well) and incubated at 37 °C for 24 h. Different CPN solutions (CPN-PSPEG-CL4, CPN-PSPEG-SCR, CPN-PSPEG, CPN-PSMA-CL4, CPN-PSMA-SCR, CPN-PSMA-sTN58 and CPN-PSMA-sTN29, 2 and 6 mg/L) were added to cells and incubated for short periods of time in absence of light (30 and 60 min).

Particle incorporation was analyzed using a BD Accuri™ C6 flow cytometer (BD Biosciences, San Jose, CA, USA). Briefly, medium containing CPNs was removed, and cells were washed twice with 500 μL of Dulbecco’s phosphate buffered saline (DPBS) and detached with trypsin and resuspended in DPBS. CNP fluorescence intensity was monitored in the green detector channel (525/30 nm) after blue light excitation (488 nm). A total of 10,000 events were analyzed for each sample, and forward scatter area in linear scale (FSC-ALin) vs. forward scatter height in linear scale (FSC-HLin) dot plot graph was used to distinguish single cells. The geometric mean fluorescence intensity value in the green channel (CPN fluorescence intensity) at the single-cell level (after doublet cells discrimination) was calculated and compared among aptamer-decorated CPN treatment and cell lines using FlowJo software (version 10.0.7).

### 2.8. In Vitro Uptake Analysis by Confocal Microscopy

MDA-MB-231, BT-474 and MDA-MB-231/cis (1.0 × 10^5^ cells/well in 24-well) were seeded on glass coverslips placed within 35 mm culture dishes and then incubated overnight in 0.5 mL of medium containing 10% FBS at 37 °C to 70–80% confluency. Afterward, the media was removed, and cells were washed with DPBS and incubated with 0.5 mL of serum-free medium containing 2 mg/L aptamer-decorated CPNs and unconjugated CPN (CPN-PSMA-CL4, CPN-PSMA-SCR, CPN-PSMA-sTN58, CPN-PSMA-sTN29 and CPN-PSMA). In all the assays, cells were incubated with aptamer-decorated CPNs diluted to the desired concentration in serum-free medium with 0.1 mg/mL yeast tRNA and 0.1 mg/mL ultrapure™ salmon sperm DNA (Invitrogen, Carlsbad, CA, USA), as non-specific competitors [28].

After incubation and three washes with DPBS, cells were fixed with 4% paraformaldehyde in DPBS for 20 min. Then, cells were incubated with conjugated wheat germ agglutinin (WGA) Alexa Fluor 647 (Invitrogen, Carlsbad, CA, USA) for 20 min at RT and washed three times with DPBS. Finally, nuclei were stained with 1.5 µM 4′,6-diamidino-2-phenylindole (DAPI, D9542, Sigma-Aldrich) in DPBS for 5 min and coverslips were mounted with glycerol/DPBS over slides [27]. For internalization experiments, MDA-MB-231/cis cells were incubated with LysoTracker Red DND-99 1:1,000 (Invitrogen, Carlsbad, CA, USA) in RPMI-1640 medium supplemented with 10% FBS previous CPN incubation. Samples were visualized by Zeiss LSM 700 META confocal microscopy (Carl Zeiss Inc., Germany) equipped with a Plan-Apochromat 63x/1.4 Oil DIC objective. Confocal fluorescence images of cells with selective detection of CPNs (green emission), cell membranes (WGA, red emission), lysosomes (LysoTracker Red, red emission) and nuclei (DAPI, cyan emission) were overlaid (merged) using ImageJ (NHI) software.

### 2.9. Subcellular Localization Assays/Traffic Endosomal Assay

For internalization experiments and colocalization with endosomal compartments, MDA-MB-231/cis cells (1.0 × 10^5^ cells/well in 24-well) were seeded on glass coverslips placed within 35 mm culture dishes (24-well plates) and incubated overnight with medium supplemented with 10% FBS before experiment. Afterwards, cells were treated with different aptamer-decorated CPNs and unconjugated CPNs (10 mg/L, CPN-PSMA-CL4, CPN-PSMA-SCR, CPN-PSMA-sTN58, CPN-PSMA-sTN29 and CPN-PSMA) for 30 min. Then, the medium containing CPNs was removed and replaced with fresh medium. Some coverslips were fixed with 4% paraformaldehyde immediately and others after 120 min. The coverslips were washed three times in DPBS and then permeabilized with PBS, 0.5% Triton X-100 for 15 min at RT before blocking in BlockAid™ blocking solution (Invitrogen, Carlsbad, CA, USA) for 30 min. Cells were incubated with mouse anti-LAMP2 and rabbit anti-EEA1 (Abcam, Cambridge, MA, USA) diluted in DPBS for 1 h at 37 °C. Coverslips were washed three-times with DPBS and treated with Alexa Fluor 568 Goat Anti-Rabbit IgG (H+L) (Invitrogen, Carlsbad, CA, USA) and Alexa Fluor 647 Goat Anti-Mouse IgG for 30 min at 37 °C. Afterwards, coverslips were washed, stained with DAPI and then mounted over glasses to visualize by confocal microscopy [32].

### 2.10. Dark Cytotoxicity

Cytotoxicity in dark condition of aptamer-decorated CPNs was evaluated by MTT and Trypan blue dye exclusion assays. To this purpose, MDA-MB-231, BT-549, BT-474 and MDA-MB-231/cis cell lines were seeded into 96-well microplates at a concentration of 10^5^ cells/mL. The next day, the culture medium was replaced with fresh medium supplemented with 10% FBS having different aptamer-decorated CPN concentrations (3, 5, and 10 mg/L) and samples were incubated for 24 h with 100 μL of tested suspensions. Later, CPN suspensions were removed and 100 μL of MTT solution (0.5 mg/mL in culture medium) was added and cells were incubated for 2 h at 37 °C in 5% CO_2_ prior to the analysis. Thereafter, the medium was removed and 100 μL of DMSO was added to dissolve blue formazan crystals. The absorbance of the formed dye was measured at 590 nm using a microplate reader. Absorbance values for untreated wells were taken as control (100% survival) [18]. For Trypan blue dye exclusion assay, cells were detached with trypsin solution after incubation for 24 h with CPN solutions (3, 5, and 10 mg/L), and then the unstained (viable) and stained (nonviable) cells were counted separately in the hemacytometer after staining with 0.4% Trypan blue solution [33]. The percentage of viable cells was calculated as follows:(1)$$\mathrm{viable}\;\mathrm{cells}\;\left(\%\right)=\frac{\mathrm{total}\;\mathrm{number}\;\mathrm{of}\;\mathrm{viable}\;\mathrm{cells}\;\mathrm{per}\;\mathrm{mL}\;\mathrm{of}\;\mathrm{aliquot}}{\mathrm{total}\;\mathrm{number}\;\mathrm{of}\;\mathrm{cells}\;\mathrm{per}\;\mathrm{mL}\;\mathrm{of}\;\mathrm{aliquot}}\times 100$$

### 2.11. In Vitro PDT Efficacy Evaluation

To evaluate the PDT effect of aptamer-decorated CPNs, breast cancer cell lines were seeded into 96-well microplates at a concentration of 10^5^ cells/mL, and after an overnight period, cells were incubated with different concentrations of aptamer-decorated CPNs (3, 5 and 10 mg/L) in serum-free DMEM for 30 min. After removing the medium, cells were washed twice with DPBS to eliminate not incorporated nanoparticles and fresh medium was added. Later, the 96-well plate was illuminated with a 96 panel LED (420 ± 17 nm) having an irradiance (radiant flux density) of 50 mW/cm^2^ (at the sample plane) for 3 min (light dose 10 J/cm^2^). Cell viability was evaluated 24 h after PDT treatment by MTT assay [18]. Three independent PDT experiments were performed (with *n* = 6 for each experiment).

### 2.12. Cell Apoptosis Analysis by Flow Cytometry

The Annexin V FLUOS staining kit (Roche Diagnostics GmbH–Mannheim, Germany) was used to assess cell apoptosis after PDT treatment according to the manufacturer’s instructions. Briefly, MDA-MB-231/cis cells were seeded into 24-well microplates at a concentration of 2 × 10^5^ cells/mL. After 24 h incubation, the medium was replaced by fresh medium supplemented with different aptamer-decorated CPN solutions (CPN-PSMA-CL4, CPN-PSMA-SCR, CPN-PSMA-sTN58 and CPN-PSMA-sTN29, 5 mg/L FBBT) for 30 min. Then, medium containing CPNs was replaced with fresh medium and the cells were exposed to light irradiation (50 mW/cm^2^ for 3 min, 10 J/cm^2^) and further cultured in an incubator for 6 and 24 h. Finally, the supernatant (floating apoptotic cells) was collected, and the adherent cells were harvested, and washed with 1x annexin-binding buffer. After centrifugation, the cells were resuspended in 1x annexin-binding buffer and 2 μL of Annexin (1 mg/mL) was added to each 100 μL cell suspension, followed by incubation at 37 °C in an atmosphere of 5% CO_2_ for 30 min. After incubation, 400 μL of 1× annexin-binding buffer was added into the samples with gentle mixing, and the samples were kept on ice before the analysis. Cells were analyzed by flow cytometry, measuring the fluorescence emission at 530 (excited by 488 nm) using a BD Accuri™ C6 flow cytometer.

### 2.13. Statistical Analysis

Data were analyzed using GraphPad Prism Software, version 8.412 (GraphPad Software, La Jolla, CA, USA), and presented as mean ± standard error of the mean (SEM). One-way ANOVA and t-student analysis were applied to indicate statistical significance * *p* < 0.05; ** *p* < 0.001.

## 3. Results

### 3.1. Preparation and Characterizations of Aptamer-Decorated CPNs

We prepared CPNs by using the CP F8BT due to its exceptional features as a donor antenna to collect excitation energy and funnel it towards molecular dopant PS acceptors, such as the porphyrin PtOEP [16,18,19]. Two functional polystyrene-based polymers (PS-PEG-COOH and PSMA) were chosen and incorporated in ~20% mass ratio to F8BT mass in order to improve colloidal stability and introduce COOH groups on the surface of CPNs [31]. The latter allowed the subsequent formation of amide bond coupling between carboxylic from functionalized CPNs (CPN-PSPEG or CPN-PSMA) and amine from NH_2_-modified aptamers by performing an EDC binding reaction (Figure 1). Using the nanoprecipitation method, functionalized CPNs with both polystyrene-based polymers produced stable nanoparticle suspensions that can be stored for several months without signs of precipitation or agglomeration.

The particle size and zeta potential of CPNs were measured before and after the EDC reaction and aptamer decoration by dynamic light scattering (Figure 2a,b). The results show a similar particle size distribution of functionalized CPNs (CPN-PSPEG and CPN-PSMA) before aptamer decoration in the range of ~20 nm. The two polystyrene-based polymers allowed us to successfully decorate the CPN surface with aptamers; however, PSMA functionalization improved the CPN performance in purification procedures and cell binding assays. The EDC reaction and aptamer surface decoration greatly increased the particle size distribution of CPN-PSPEG probably due to the agglomeration induced by the loss of surface negative charges (Table 1). It was not possible to recover aptamer-decorated CPN-PSPEG in large quantities after the membrane filtration procedure, which also indicates an increase in large-sized nanoparticles and agglomeration after aptamer conjugation. On the other hand, CPN-PSMA conserved a suitable size distribution with an expected small increase in size, and a change in zeta potential due to the aptamer conjugation. Besides, CPN-PSMA could be recovered with great success after the membrane filtration process.

Similar results related to size and zeta potential changes were obtained in gel electrophoresis assay. Bare CPNs showed more mobility through the pores of the gel than EDC-activated or aptamer-decorated CPNs (Figure 2c). On the other hand, when EDC-activated CPNs were compared against aptamer-decorated CPNs, the mobility of nanoparticles into the gel was also different, which is in agreement with DLS experiments (Appendix A). This result also confirms the successful binding of aptamers into the CPN surface for both types of functionalized CPNs (CPN-PSPEG and CPN-PSMA).

Figure 2d (upper panel) shows the normalized absorption and emission spectra of a series of CPNs without PtOEP. The absorption spectra of CPNs decorated with aptamers show increased absorption around 255–260 nm, which is associated to the presence of aptamers [34,35]. The shape and intensity of emission spectra is identical for all samples.

The doping concentration of PtOEP was 5 mg/L (~10 wt% relative to F8BT polymer), which was recalculated by comparing absorption spectra before and after water injection. In previous reports, we established that the incorporation of porphyrin dopants into the CPN matrix is essentially quantitative, based on a combination of several experiments: absorption and emission measurements, oxygen consumption/singlet oxygen generation, and arguments based on the hydrophobicity of the tetrapyrrole (resulting in null solubility in aqueous media) [16,18]. Figure 2d (lower panel) shows that kinetic traces of ADPA consumption (ADPA-O_2_ formation) upon selective irradiation of CPN-PSPEG and CPN-PSMA samples are essentially identical. The insert shows selected spectra at different irradiation times for experiments using CPN-PSMA. Absorption bands with peaks at ~460 and 537 nm corresponding to F8BT and PtOEP remain constant, demonstrating the good photostability of the CPN photosensitizers. Analogous experiments using aptamer-decorated CPN samples show similar results. Overall, these results suggest that the ^1^O_2_ quantum yields for all prepared CPN samples are the same and equal to the value reported for CPN-PSPEG (0.24) [16]. Although this work was not aimed at establishing all the possible ROS produced by the particles, we decided to study the formation of ^1^O_2_, a highly relevant reactive oxygen species. It is important to note that other ROS species might be forming and playing a role in the observed PDT effect.

### 3.2. TNBC Targeting Using Aptamer-Decorated CPNs Measured by Flow Cytometry

To demonstrate the selective binding of aptamer-decorated CPNs to TNBC cells, we incubated MDA-MB-231 and BT-549 cells with different aptamer-decorated CPNs for short periods (30 and 60 min), and evaluated green fluorescence emission coming from cells taking advantage of the intrinsic high fluorescence of the CPNs prepared without PtOEP, and the profuse number of cells evaluated by this technique. Figure 3 shows single-cell emission histograms for EGFR-positive MDA-MB-231 and BT-549 before (control, red) and after incubation (30 min) with CPN-PSPEG or CPN-PSMA conjugated with well-established CL4 aptamer (able to bind at high affinity to EGFR) and SCR as an unspecific negative control [36]. Cytometry experiments performed after CPN (6 mg/L) incubation for 30 min with EGFR-positive TNBC cell lines indicate that CPN-PSMA were able to selectively bind to MDA-MB-231 (Figure 3a) and BT-549 cells (Figure 3b) when they are decorated with CL4 aptamer vs. SCR aptamer and CPNs without aptamer conjugation. The percentages of green fluorescence cell population in MDA-MB-231 cells at 30 min incubation time with CPNs were 59 ± 5, 61 ± 2, and 95 ± 3, for CPN-PSMA, CPN-PSMA-SCR and CPN-PSMA-CL4 respectively, while for CPN-PSPEG, CPN-PSPEG-SCR and CPN-PSPEG-CL4 the percentages were 50 ± 1, 45 ± 5, and 58 ± 3, respectively. CPN-PSMA were able to label TNBC cells in a more pronounced manner compared to CPN-PSPEG, and this difference was superior when CL4-decorated CPN-PSMA were employed. In EGFR-positive TNBC BT-549, the behavior was similar. It is worth noting that the binding ability of CPN-PSMA-CL4 to TNBC is significantly higher than that of CPN-PSPEG-CL4. This difference could be rationalized by considering, among others, the following aspects: (i) the degree of aptamer functionalization for each particle type could be different, resulting in different binding affinities, and (ii) the intrinsic affinity of CPN-PSMA particles towards all TNBC cell lines is higher than that of CPN-PSPEG (as seen in Figure 3). Based on these results, the following binding affinity and PDT efficacy assays were performed using PSMA as the best polystyrene-based polymer to be introduced in CPN synthesis in order to perform the further aptamer conjugation.

In order to increase the repertoire of aptamer-conjugated CPNs for TNBC targeting, we decided to continue evaluating the ability of novel anti-TNBC nuclease-resistant RNA aptamers [28,29] to promote the binding of CPNs and distinguish TNBC cells and also cisplatin-resistant derivatives from non-TNBC breast cancer cells. To accomplish this, we used NH_2_-modified sTN58 and sTN29 aptamers and conjugated them to the CPN-PSMA surface to finally assess the binding ability of aptamer-decorated CPNs by flow cytometry. To evaluate the binding ability of CPNs against breast cancer cells, the green fluorescence emission counts at the single-cell level obtained by flow cytometry were used to compare among the different aptamer-decorated CPNs at the same concentration tested [19]. Bar graphs from each TNBC and non-TNBC cell line (Figure 4b–d) represent the geometric mean fluorescence intensity in the green channel of three independent experiments obtained by flow cytometry after the incubation of cells with CPNs 2 mg/L (Figure 4a). Emission observed in cells not incubated with CPNs (control) is ascribed to autofluorescence. Data indicate that TNBC cell lines (MDA-MB-231, BT-549 and MDA-MB-231/cis) have significantly higher particle uptake compared with the TPBC BT-474 epithelial breast cancer cell line (ER+, PR+, HER2 over-expression) at similar CPN concentration and incubation time. These positive binding results were obtained for CPN-PSMA-CL4, CPN-PSMA-sTN58 and CPN-PSMA-sTN29 with some difference among them, which likely reflect the different cell binding affinity of the unconjugated aptamers [28,29]. Interestingly, a higher and lower uptake into MDA-MB-231/cis compared to MDA-MB-231 cells was observed with CPN-PSMA-sTN58 and CPN-PSMA-sTN29, respectively, which is in agreement with the targeting behavior of the two nude aptamers vs. chemo-resistant cells and the parental counterpart [28,29]. Nanoparticles conjugated with non-specific SCR aptamer had similar behavior to non-decorated CPN, suggesting that the cell uptake that incubated cells with these CPNs was assigned to an unspecified nanoparticle–cell interaction due to the fact that a portion of the CPN surface is still hydrophobic, which causes non-specific labeling [31]. This unspecified interaction decreased when other non-specific competitors were used in the incubation procedure, such as yeast tRNA and salmon sperm DNA. These results were confirmed using the non-TNBC cell model (BT-474), in which all aptamer-decorated CPN-PSMA and non-decorated CPNs had similar cell fluorescence intensity signal values, which were lower than in the TNBC cells, where the developed aptamers were originally Cell-SELEX-selected (Figure 4d).

### 3.3. Uptake Analysis of Aptamer-Decorated CPNs by Confocal Microscopy

To assess whether the proposed aptamers are able to specifically target CPNs to TNBC cells and enhance their intracellular uptake, we used very low CPN concentration for the incubation process and unspecific competitors in order to block non-specific interaction between CPN and cells. To investigate the intracellular localization of aptamer-decorated CPNs, we took advantage of their intrinsic fluorescent properties. Particle distribution and colocalization with cell organelles were performed using confocal fluorescence microscopy. As shown in Figure 5, the signal associated with aptamer-decorated CPNs was clearly visible in the cell membrane and cytoplasm of MDA-MB-231/cis cells at 30 min and further increased in a time-dependent manner (60 min). Conversely, a very weak signal was detected with SCR-decorated CPN and unconjugated CPNs as well. These findings were reproduced in the other TNBC cell lines MDA-MB-231 and BT-549 (Appendix A). Moreover, the green signal associated with all the different aptamer-decorated CPNs tested was insignificant in non-TNBC BT-474 cells (Appendix A). The confocal microscopy results were in agreement with flow cytometry results. sTN58 aptamer greatly enhanced the CPN incorporation into cisplatin-resistant MDA-MB-231 cells (Figure 5a). Based on colocalization analysis, aptamer-decorated CPNs were able to follow the traffic kinetic of aptamers with an intracellular destination at lysosomes (Figure 5b).

Altogether, these data clearly indicate that CL4, sTN29 and sTN58 aptamers specifically deliver CPN to TNBC cells, strongly enhancing their intracellular uptake.

In order to confirm cell internalization kinetic for aptamer-decorated CPNs, we performed an endosomal traffic assay incubating MDA-MB-231/cis cells with different aptamer-decorated CPNs for only 30 min and then evaluated colocalization with endosomal compartments up to 120 min (Figure 6). The results show a colocalization of aptamer-decorated CPNs, mainly sTN58-decorated CPNs, with early endosomes (EEA1 positive-marked organelles) at 30 min (Pearson coefficient index (PCI) of 0.54 ± 0.1 for EEA1 and CPN-PSMA-sTN58 vs. PCI of 0.2 ± 0.1 for LAMP2 and CPN-PSMA-sTN58); meanwhile, the colocalization of sTN58-decorated CPNs with late endosomes/lysomes (LAMP2 positive-marked organelles) was superior at 120 min post-incubation time (PCI of 0.23 ± 0.1 for EEA1 and CPN-PSMA-sTN58 vs. PCI of 0.5 ± 0.2 for LAMP2 and CPN-PSMA-sTN58).

### 3.4. Cytotoxicity of Aptamer-Decorated CPNs in Dark Condition

The biocompatibility of new aptamer-decorated CPNs with TNBC cells was assayed using a mitochondrial functionality assay (MTT), and a dye exclusion assay that evaluates cell membrane integrity (Trypan blue exclusion). Figure 7 shows the results from TNBC and non-TNBC cells exposed to increasing aptamer-decorated CPN concentrations up to 24 h. Very limited basal toxicity of aptamer-decorated CPNs without light exposure was observed up to 10 mg/L CPNs after a 24 h incubation period, but this was not statistically different from the mock control group (Figure 7). These data confirm that aptamer-decorated CPNs are extremely safe, at least in the concentrations tested in PDT experiments. Besides, the safety behavior was confirmed using two different cell viability assays and also in different breast cancer cells (TNBC and non-TNBC cells).

### 3.5. Photodynamic Evaluation of Aptamer-Decorated CPNs in TNBC Cells

In order to assess the ability of aptamers to improved CPN-PDT efficacy, cells were incubated for 30 min with various concentrations of nanoparticles, ranging from 3 to 10 mg/L F8BT, then irradiated with blue light for 3 min (10 J/cm^2^) and cell viability was determined 24 h post PDT using MTT assay and apoptosis flow cytometry determination. In the cisplatin-resistant TNBC cell model (MDA-MB-231/cis), different aptamers selected to decorate CPNs were able to improve the PDT cytotoxicity of our doped nanoparticles. The cell viability of MDA-MB-231/cis cells was superior to 90% when unconjugated CPNs were used with a non-cytotoxic light irradiance dose (10 J/cm^2^) (Appendix A) in this shorter incubation time (30 min) (Figure 8a). At the same CPN concentrations, cell viability decreased significantly for all specific aptamer-decorated CPNs used, with a particular interest in sTN58-decorated CPN, where cell viability was diminished up to 53.6 ± 13, 37 ± 10.5 and 23.9 ± 3.8 for 3, 5 and 10 mg/L CPN, respectively, with a 10 J/cm^2^ light dose. These data clearly show the concentration-dependent cytotoxic effect of nanoparticles when specific aptamers for these chemotherapeutic-resistant cells are used as target moieties. It is noteworthy that CPNs decorated with the non-specific SCR aptamer had similar behavior to bared nanoparticles. In the parental cells MDA-MB-231, CL4 was the most promising aptamer in improving PDT efficacy compared with sTN58 and sTN29 (Appendix A) and clearly reduced cell viability at the higher concentration tested (viability of 50.8 ± 2% at 10 mg/L) compared with unconjugated CPN-PSMA (77 ± 6% at 10 mg/L) and CPN-PSMA-SCR (83 ± 2% at 10 mg/L). These results are in agreement with flow cytometry incorporation experiments, where different aptamer-decorated CPN-PSMA incorporated to a greater extent in the resistant cell line compared to parental MDA-MB231, suggesting that the PDT cytotoxic effect is due to a greater cellular incorporation of nanoparticles with a similar CPN concentration incubation, time exposure and light dose used.

The CPN-PDT-induced mechanism of cell death induced by CPN-PDT previously reported by our group [18] was investigated in order to demonstrate the ability of aptamer-decorated CPNs to achieve apoptotic cell death in lower CPN concentrations and exposure time. To this end, MDA-MB-231/cis cells were incubated with equal aptamer-decorated CPN concentration for 30 min and irradiated with blue light (10 J/cm^2^) after nanoparticle removal. As shown in Figure 8c, the incubation of cells with CPN-PSMA without aptamer decoration did not induce cell apoptosis after light irradiation (live cells >85%). Additional control experiments with cells without particle incubation and exposed to equal light dose (negative control) showed a lived cell percentage >85%, while exposed cells to a cytotoxic light dose (100 J/cm^2^) showed an apoptotic cell percentage of 100% and were used as a positive control of the technique (Figure 8b). The percentage of apoptotic cells varies depending on the aptamers chosen to decorate the CPNs. Nanoparticles decorated with CL4 and sTN58 were the most significant inducers of apoptosis at equal CPN concentration after 6 h PDT, showing annexin V positive percentages of 48 ± 2 and 61 ± 2, respectively (Figure 8c).

## 4. Discussion

The development of new drug delivery systems that promote optimal therapeutic efficiency allied with low side effects is a challenge in the biomedical field. Nanomedicine brings numerous innovative materials to develop multifunctional nanosystems for the diagnosis and treatment of several types of cancer and is revolutionizing the delivery and enhancing the effectiveness of biologically active molecules [6]. This postulation is valid when biological moieties are identified for specific tumor recognition, and in our case the development of selective TNBC targeting agents would greatly advance the development of personalized therapy for the most aggressive type of breast cancer, and where the specific treatments employed for other subtypes of breast cancer have no place. In this sense, a panel of 2’F-Pys-RNA aptamers that bind with high affinity to TNBC cells and had inherent antiproliferative action have been developed previously [28,37], and could serve to create novel targeted treatment approaches able to eliminate each individual TNBC subtype, diminish toxicity or side effects by reducing drug concentration used, and overcome the resistance to chemotherapy [27]. For these reasons, aptamers have received increasing attention for cancer diagnosis and therapy, as a result of their low molecular weight, low/lack of immunogenicity, and versatility in manipulation for improved stability and targeting efficacy of conventional and experimental therapeutics [28,36,38]. All these characteristics inherent to aptamers are sought to be added in the rational design of theranostic nanoparticles for cancer management [6,39]. In the present work, we proposed to develop CP nanoparticles decorated with specific and novel aptamers able to bind to TNBC cells in order to improve the efficacy of PDT, which is in constant examination preclinically and clinically to treat primary breast tumors [8]. Different aptamers were used in the past to increase the selectivity of various types of PSs [40]; however, to our knowledge this is the first time that RNA aptamers (with known and not yet identified TNBC target receptors) have been conjugated to the surface of CP nanoparticles and evaluated in TNBC cells for labelling and therapeutic purposes.

In order to tip up aptamers to the CPN surface, we used modified NH_2_-terminated aptamers, and two functional amphiphilic polystyrene-based polymers (PS-PEG-COOH and PSMA) incorporated in a ~20% mass ratio in the synthesis of CNPs. The comb-like PS-PEG-COOH was previously employed by our group with the aim of increasing the colloidal stability of the resulting nanoparticles for biological experiments [18,22], and in this opportunity the carboxyl groups added to the surface of CPNs will subsequently allow for the formation of the amide bond through EDC coupling with the proposed aptamers. Alternatively, polystyrene-based polymer PSMA, which is also included in the same mass ratio into CPNs, is hydrolyzed in the aqueous environment after CPN synthesis to generate carboxyl groups in the surface for the further EDC coupling reaction. PSMA, a biocompatible copolymer, was previously used for the successful conjugation of different molecules for medical applications, and with pronounced improvements in pharmacological properties of the resulting nanoparticles [41,42,43]. A similar approach was employed by Wang et al. to successfully bind antibodies to the CPN surface for cell labelling purposes with an improvement in the labelling of intracellular structures when PSMA was included in CPNs [31]. In our experiments, the incorporation of PS-PEG-COOH or PSMA in the formulations has shown a differential impact on the size of the unconjugated nanoparticle and it could be associated with the extended particle solvation shell consisting of water molecules strongly interacting with extended PEG-COOH chains in CPN-PSPEG, which is not presented in CPN-PSMA [18]. The expected increment in size by the incorporation of aptamers into the CPN surface was more manifest for CPN-PSPEG, which did not necessarily reflect a greater accumulation of aptamers on the surface of the nanoparticles, and therefore the performance of these nanoparticles in cell labelling was lower compared to CPN-PSMA. On the other hand, the presence of COOH^−^ groups on the CPN-PSPEG and CPN-PSMA surface resulted in a strong negative surface charge, which might provide advantages such as decreasing the non-specific interaction between the negative charged aptamers and CPN surface [44], thus preserving aptamer conformation and binding characteristics, and only the interaction of these biomolecules and CPN would be possible after the EDC coupling reaction. The change in the zeta potential of CPN-PSPEG and CPN-PSMA, following its modification with aptamers, further confirmed the conjugation. In our previous work, PLGA-based nanovectors had similar behavior regarding zeta potential and size changes after aptamer decoration [27]. Based on the characterization and binding affinity evaluation of the two nanoformulations proposed, PSMA was the most promising stabilizer polymer to develop a nanoplatform based on CPs, molecular dopants and aptamers to selectively recognize TNBC cells. One of the differences between PSMA and PS-PEG-COOH, in the same mass ratio used, is that PSMA brings a higher amount of available COOH than PS-PEG-COOH for the further aptamer functionalization.

In an attempt to explore if improved photodamage-induced PDT activity was achieved by the aptamer decoration of CPNs, we chosen three nuclease-resistant 2′FPy-containing RNA aptamers (CL4, sTN58 and sTN29) previously validated for TNBC targeting. CL4 (39 mer) binds at high efficacy to EGFR overexpressed on the cell surface in several human cancers [26], including TNBC [27,36]. sTN58 and sTN29 (45 and 41 mer, respectively), previously selected by a TNBC cell-SELEX approach [28], bind to cytomembrane proteins of TNBC cells, distinguishing them from both normal and TPBC cells [28,29]. First, the ability to uptake and tolerate aptamer-decorated CNPs (in dark condition) in different TNBC and non-TNBC cells was evaluated. We demonstrated that CPN-PSMA decorated with CL4 aptamer was able to bind with high affinity (based on the short incubation time exposure and the lower CPN concentration employed) to TNBC MDA-MB-231 and BT-549. The anti-EGFR CL4 aptamer [36,45] not only improved the binding ability to the membrane receptor of TNBC but also favored cell internalization of CPNs compared to unconjugated CPNs and CPN decorated with SCR aptamer (CPN-PSMA-SCR). Based on these results and the recent work of our [27], and other, research groups (reviewed in [26]), CL4 aptamer endows different polymeric nanosystems with excellent cancer cell targeting, rapid uptake and internalization capabilities in EGFR-positive TNBC cells. Moreover, the efficient cell internalization improvement by CL4 aptamer was transferred into the effective cytotoxic effect after the light irradiation procedure. It is noteworthy that the phototoxic effect previously demonstrated for these nanoparticles without any active targeting was achieved in a higher concentration and longer incubation time [18,19], and here the aptamer decoration of CPNs meant an improvement in these parameters. The anti-TNBC sTN58 and sTN29 aptamers [28,29] also increased the efficacy of CPN-PDT in MDA-MB-231, which is in agreement with the flow cytometry results.

In addition, the chemoresistance of TNBC is a major reason for treatment failures, and therefore the searching for new therapeutic options is mandatory. Based on the flow cytometry binding results with CPN-PSMA decorated with the different TNBC aptamers (CL4, sTN58 and sTN29), we evaluated the performance of PDT with aptamer-decorated CPNs and demonstrated that the PDT efficacy of all the three different formulations was superior to CPN-PSMA-SCR and unconjugated CPN-PSMA in chemo-resistant MDA-MB-231/cis cells. Better results among the aptamers were obtained with TN58, which is in agreement with its higher affinity for chemo-resistant cells [28,29], and therefore it is an ideal candidate to investigate specific active targeting treatments for chemo-resistant tumors.

Taken together, our results demonstrate that CL4, sTN58 and sTN29 aptamers drive CPNs to TNBC cells with an improvement in PDT in vitro protocols. It is known that the penetration depth of light in biological tissues is highly dependent on its wavelength. Although light sources centered at 420 nm are not expected to have high penetration depths in biological tissue, the in vivo successful application of PDT protocols using light in this range has been previously reported in several articles [46,47,48]. The effective penetration of sufficient photons to elicit a reasonable phototherapeutic effect it is a complex combination of absorption, scattering, and diffraction of the target tissue and of the effective fluence of the light source. Furthermore, we envision that CPN functionalized with aptamers developed in this work could be used in PDT protocols as an adjuvant post-surgical treatment (in tissue regions readily accessible for superficial irradiation) to eliminate remaining tumor cells. Taking these observations into account, the limited penetration of blue light into tissue is not expected be an inconvenience. Nevertheless, accurate preclinical characterization to evaluate the pharmacokinetics, toxicology and in vivo tumor-specific anticancer activity of these aptamer-decorated CPNs is on-going in our laboratories and is a fundamental step toward the validation of their clinical utility.

## 5. Conclusions

We were able to prepare optimized aptamer-decorated CPNs with suitable size and colloidal stability for the PDT treatment of TNBC cells. Different aptamers employed to conjugate into the CPN surface conserved the binding activity for specific TNBC cell populations, and therefore improved the cell internalization of CPNs and their therapeutic efficacy, reducing drug exposure and the concentration used to photokill TNBC after the PDT procedure. In the constant search for the development of new biomedical nanomaterials with a theranostic purpose, aptamer-conjugated polymers could represent a new nanoplatform for the treatment of TNBC using a PTD approach.

## Figures and Tables

**Figure 1 pharmaceutics-14-00626-f001:**
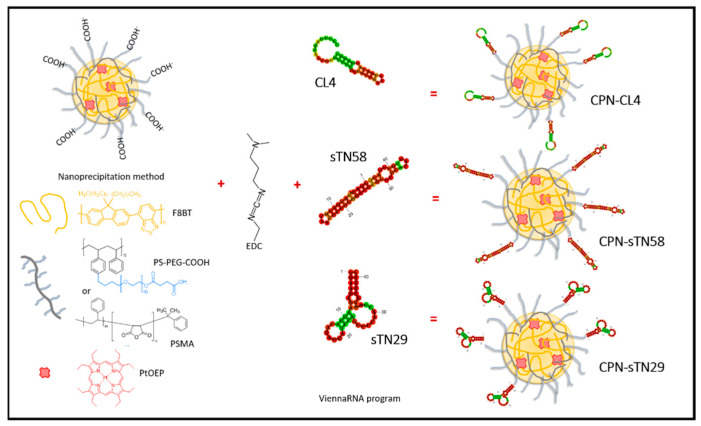
Aptamer decoration of metallated porphyrin-doped CPNs using EDC coupling reaction for photodynamic therapy of TNBC cells.

**Figure 2 pharmaceutics-14-00626-f002:**
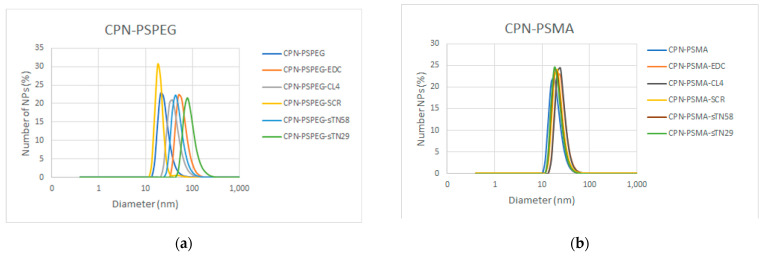
Characterization of aptamer-decorated CPNs. Dynamic light scattering data of CPN-PSPEG (**a**) and CPN-PSMA (**b**) before and after EDC COOH-activating reaction and aptamer conjugation. (**c**) Gel electrophoresis of different CPN-PSMA nanoparticles in 1.5% agarose gel. (**d**). Upper panel, absorption (solid lines, left axis) and emission (dashed lines, right axis) spectra of a different CPN without PtOEP decorated with CL4, sTN58, sTN29 and SCR aptamers. Lower panel CPNs photosensitized ADPA in the presence of air monitored at 400 nm. Insert: Absorption spectra of the ADPA solution (~4 × 10^−5^ M) in presence of CPNs (12 mg/L) at different irradiation times with a blue led.

**Figure 3 pharmaceutics-14-00626-f003:**
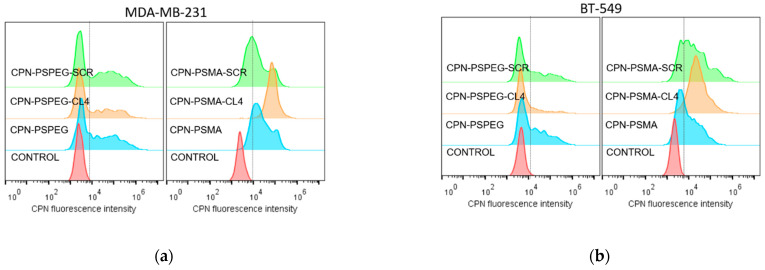
Flow cytometry assay to evaluate binding ability of aptamer-decorated CPN in TNBC cell lines. Representative single-cell emission histograms for EGFR-positive TNBC MDA-MB-231 (**a**) and BT-549 (**b**) before (control, red) and after incubation (30 min) with CPN-PSPEG or CPN-PSMA conjugated with CL4 or SCR.

**Figure 4 pharmaceutics-14-00626-f004:**
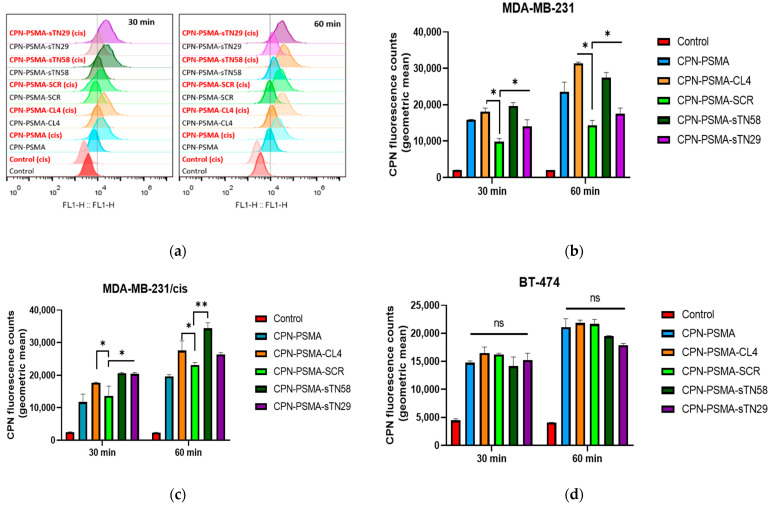
Uptake quantification of aptamer-decorated CPNs by flow cytometry in TNBC cells. (**a**) Representative single-cell emission peaks of MDA-MB-231 and MDA-MB-231/cis cells exposed to CPN suspension (2 mg/L) for 30 and 60 min. Geometric mean fluorescence intensity obtained for MDA-MB-231 (**b**), MDA-MB-231/cis (**c**) and BT-474 cells (**d**). Error bars represent the standard deviation of three independent experiments. * *p*-value < 0.01 and ** *p*-value < 0.001 with ANOVA test, ns: no statistically significant differences.

**Figure 5 pharmaceutics-14-00626-f005:**
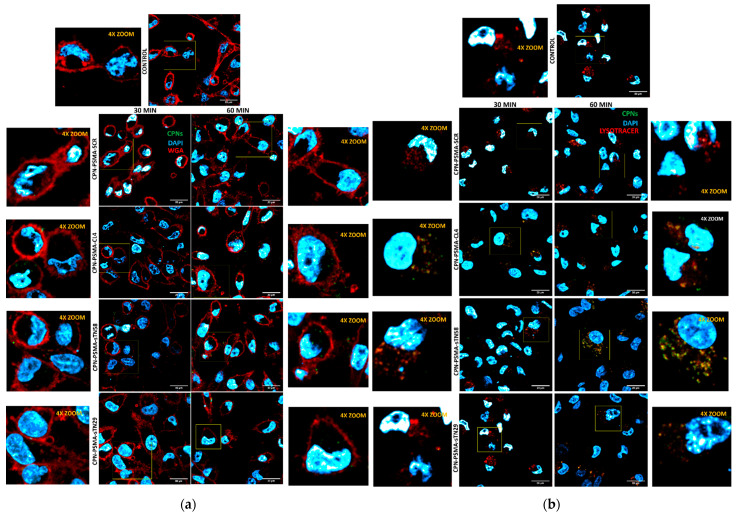
Intracellular localization of aptamer-decorated CPNs in TNBC cells. Representative confocal images of MDA-MB-231/cis cells exposed to 2 mg/L CPNs for 30 and 60 min and stained with WGA for cell membrane visualization (**a**) or LysoTracker Red for lysosome visualization (**b**).

**Figure 6 pharmaceutics-14-00626-f006:**
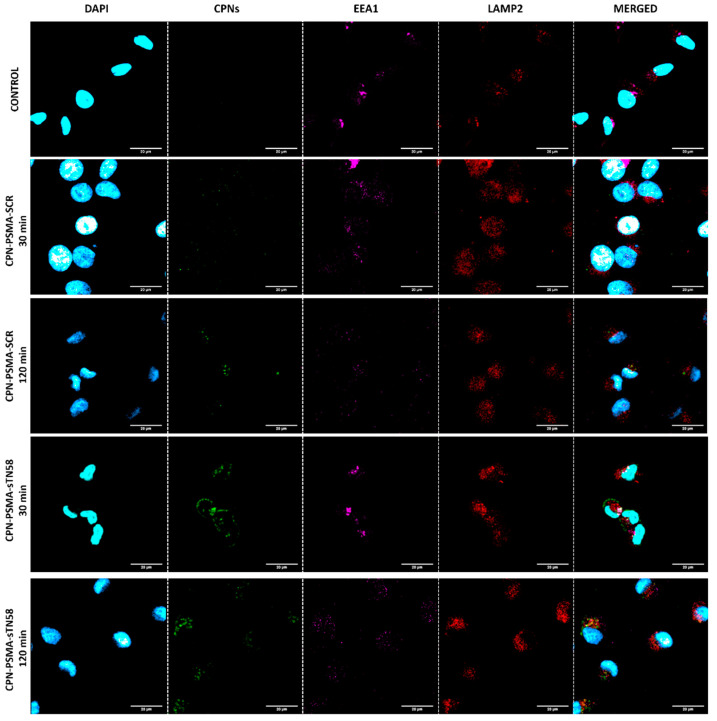
Endosomal traffic assay with aptamer-decorated CPNs in TNBC cells. Representative confocal images of MDA-MB-231/cis cells exposed to different aptamer-decorated CPNs (2 mg/L) for 30 min, and then early endosome and late endosome/lysosomes were stained at 30 and 120 min.

**Figure 7 pharmaceutics-14-00626-f007:**
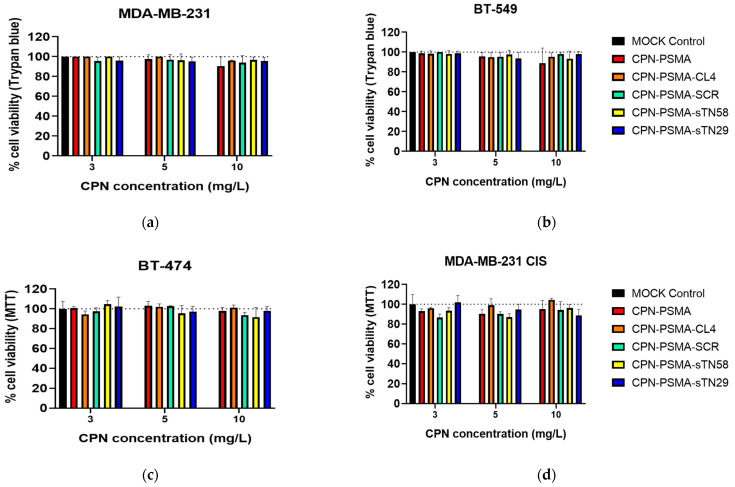
Biocompatibility of aptamer-decorated CPN-PSMA in TNBC cells. Cell viability evaluated by Trypan blue exclusion assay after 24 h incubation time of TNBC MDA-MB-231 (**a**) and BT-549 (**b**) with increased concentration of aptamer-decorated CPNs. Cell viability evaluated by MTT assay after 24 h incubation time of non-TNBC BT-474 (**c**) and TNBC MDA-MB-231/cis (**d**) with increased concentration of aptamer-decorated CPNs.

**Figure 8 pharmaceutics-14-00626-f008:**
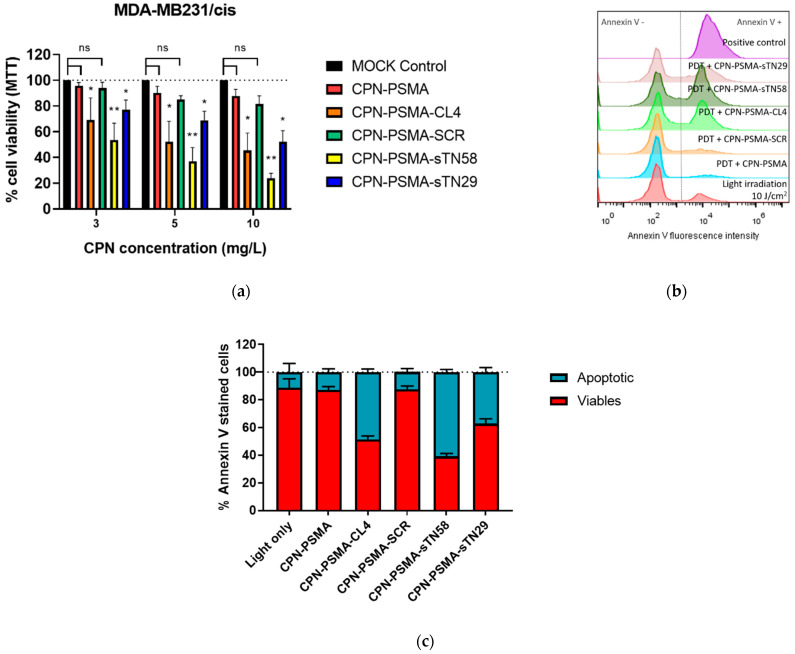
Photodynamic evaluation of aptamer-decorated CPN-PSMA in TNBC cells. (**a**) Cell viability (evaluated by MTT) after 24 h PDT treatment of MDA-MB-231/cis cell exposed to increased concentration of aptamer-decorated CPNs for 30 min and irradiated with blue light (10 J/cm^2^). (**b**) Representative flow cytometry histograms corresponding to apoptosis analysis after PDT treatment (annexin V staining) with aptamer-decorated CPN-PSMA after 6 h post PDT (CPN concentration of 5 mg/L for 30 min and a light dose of 10 J/cm^2^). (**c**) Bar graph of percentages of viable and apoptotic cells obtained from experiments described in (**b**). * p-value < 0.01 and ** p-value < 0.001 with ANOVA test, ns: no statistically significant differences.

**Table 1 pharmaceutics-14-00626-t001:** Hydrodynamic diameter and zeta potential of different aptamer-decorated CPNs and nonconjugated CPNs.

CPNs	Diameter (nm)	Zeta Potential (mV)
CPN-PSPEG	26 ± 8	−33 ± 9
CPN-PSPEG-CL4	42 ± 15	−8 ± 7
CPN-PSPEG-SCR	36 ± 10	−3 ± 5
CPN-PSPEG-sTN58	51 ± 18	−12 ± 7
CPN-PSPEG-sTN29	98 ± 20	−27 ± 10
CPN-PSMA	19 ± 7	−45 ± 19
CPN-PSMA-CL4	22 ± 6	−18 ± 13
CPN-PSMA-SCR	23 ± 7	−35 ± 17
CPN-PSMA-sTN58	21 ± 6	−25 ± 8
CPN-PSMA-sTN29	21 ± 6	−14 ± 5

## Data Availability

Not applicable.

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
