# Peer review of "Selective Photo-Assisted Eradication of Triple-Negative Breast Cancer Cells through Aptamer Decoration of Doped Conjugated Polymer Nanoparticles"

_pharmaceutics, 2022, doi:10.3390/pharmaceutics14030626_

Round 1
Reviewer 1 Report
In this work, the authors reported optimized aptamer-decorated CPNs for PDT treatment of TNBC cells. It was confirmed that surface conjugation of aptamers could enhance the specificity of CPNs for PDT of TNBC. Some specific issues should be addressed before further considerations.
- What`s the difference of the aptamers between CL4, sTN58, sTN29?
- The absorption of CPN-PSPEG and CPN-PSMA was around 500 nm. The limitation of the light penetration for PDT against should be discussed.
- The loading amounts of porphyrin should be evaluated.
- In Figure 3, CPN-PSMA-CL4 could dramatically enhance the cell binding ability of TNBC than that of CPN-PSPEG-LC4. Why?
Author Response
Reviewer 1
In this work, the authors reported optimized aptamer-decorated CPNs for PDT treatment of TNBC cells. It was confirmed that surface conjugation of aptamers could enhance the specificity of CPNs for PDT of TNBC. Some specific issues should be addressed before further considerations.
1. What`s the difference of the aptamers between CL4, sTN58, sTN29?
We thank the Reviewer for giving us the opportunity to better clarify this issue. CL4, sTN58 and sTN29 are RNA aptamers, containing 2’FPys all along the sequence, which were chosen for nanoparticle functionalization due to their previous validated TNBC cell targeting. The nucleotide sequence of the three aptamers is reported in the Materials and Methods Section. We have better clarified the difference among the three aptamers in the discussion of the revised manuscript.
Paragraph added: we chose three nuclease-resistant 2’FPy-containing RNA aptamers (CL4, sTN58 and sTN29) previously validated for TNBC targeting. CL4 (39 mer) binds at high efficacy to EGFR overexpressed on cell surface in several human cancers (26), including TNBC (27, 36). sTN58 and sTN29 (45 and 41 mer, respectively), previously selected by a TNBC cell-SELEX approach (28), bind to cytomembrane proteins of TNBC cells discriminating them from both normal and TPBC cell (28,29).
2. The absorption of CPN-PSPEG and CPN-PSMA was around 500 nm. The limitation of the light penetration for PDT against should be discussed.
We agree with the reviewer's comment, a paragraph regarding this issue was included in the discussion section of the revised manuscript.
Paragraph added: It is known that the penetration depth of light in biological tissues is highly dependent on its wavelength. Although light sources centered at 420 nm are not expected to have high penetration depths in biological tissue, the in vivo successful application of PDT protocols using light in this range has been previously reported in several articles [46–48]. The effective penetration of sufficient photons to elicit a reasonable phototherapeutic effect it is a complex combination of absorption, scattering, and diffraction of the target tissue and of the effective fluence of the light source. Furthermore, we envision that CPN functionalized with aptamers developed in this work could be used in PDT protocols as an adjuvant post-surgical treatment (in tissue regions readily accessible for superficial irradiation) to eliminate remaining tumor cells. Taking these observations into account, the limited penetration of blue light into tissue is not expected be an inconvenience.
3. The loading amounts of porphyrin should be evaluated.
We thank the reviewer's for bringing up this point. We added the following sentences in the results section to clarify this issue.
Paragraph added: The doping concentration of PtOEP was 5 mg/L (∼10 wt % relative to F8BT polymer), which was recalculated by comparing absorption spectra before and after water injection. In previous reports we established that the incorporation of porphyrin dopants into the CPN matrix is essentially quantitative, based on a combination of several experiments: absorption and emission measurements, oxygen consumption/singlet oxygen generation, and arguments based on the hydrophobicity of the tetrapyrrole (resulting in null solubility in aqueous media) [16,18].
4. In Figure 3, CPN-PSMA-CL4 could dramatically enhance the cell binding ability of TNBC than that of CPN-PSPEG-LC4. Why?
We appreciate the reviewer's comment. The following sentences have been added into the result section to address this issue.
Paragraph added: It is worth noting that the binding ability of CPN-PSMA-CL4 to TNBC is significantly higher than that of CPN-PSPEG-CL4. This difference could be rationalized by considering, among others, the following aspects: i) the degree of aptamer functionalization for each particle type could be different resulting in different binding affinities, and ii) the intrinsic affinity of CPN-PSMA particles towards all TNBC cell lines is higher than that of CPN-PSPEG (as seen in Fig 3).
Reviewer 2 Report
The manuscript by Luis Exequiel Ibarra et al. entitled Selective photo-assisted eradication of Triple-Negative breast cancer cells through aptamer decoration of doped conjugated polymer nanoparticles describes the preparation of photoactive materials which were assessed in vitro for their potential usefulness in anticancer PDT. The studied topic is important, as cancer still is the leading cause of death in developed countries, and PDT seems to be a promising new therapeutic option.
The study is very well designed and the manuscript good written. The biological evaluation studies are impressive. The paper fits the scope of journal Pharmaceutics.
My comments:
- Characterization of the materials. Currently, there are no studies presented that would prove that the materials were conjugated with the aptamers, apart from such indirect indications as change in the mobility in electrophoresis and change in the biological activity. Please provide some results that would show the presence of aptamers on the surface (i.e. FTIR, TG-DSC, XPS) and enable to assess whether the prepared bathes of the material indeed do not differ significantly. Also, it would be nice to see an electron microscope micrograph of the material to see the morphology of the nanoparticles.
- I noticed that blue light (420 nm) was used for irradiation. This greatly hampers the potential use of this material in studies other than in vitro, due to the low penetration of light of such low wavelength through tissues. Limitations of the study should be discussed and possible solution proposed. Other than that, I did not find the information whether the effect of light alone on cells was tested, without the photosensitizer?
- You tested the mediation of singlet oxygen generation, but did you consider that other ROS may be produced, such as for example hydroxyl radical?
- Typos/English/style: “in vitro e in vivo” (line 88); “2’Fluoro-pyrimidines” – should be 2’-fluoropyrimidines (129); “O-acylisourea” – O should be italicized (156); “4’,6-Diamidino-2- phenylindole” - 4’,6-diamidino-2-phenylindole (238); “citotoxicity” (482); commas are given instead of points as the decimal separator (subchapter 3.5).
Author Response
Reviewer 2
The manuscript by Luis Exequiel Ibarra et al. entitled Selective photo-assisted eradication of Triple-Negative breast cancer cells through aptamer decoration of doped conjugated polymer nanoparticles describes the preparation of photoactive materials which were assessed in vitro for their potential usefulness in anticancer PDT. The studied topic is important, as cancer still is the leading cause of death in developed countries, and PDT seems to be a promising new therapeutic option.
The study is very well designed and the manuscript good written. The biological evaluation studies are impressive. The paper fits the scope of journal Pharmaceutics.
My comments:
1. Characterization of the materials. Currently, there are no studies presented that would prove that the materials were conjugated with the aptamers, apart from such indirect indications as change in the mobility in electrophoresis and change in the biological activity. Please provide some results that would show the presence of aptamers on the surface (i.e. FTIR, TG-DSC, XPS) and enable to assess whether the prepared bathes of the material indeed do not differ significantly. Also, it would be nice to see an electron microscope micrograph of the material to see the morphology of the nanoparticles.
We thank the reviewer for his/her suggestion. We agree that the suggested techniques would be a nice addiction however, we feel that these are not simple or feasible to apply for a clear identification of the surface of our material due to the following characteristics of our nanosystems: concentrations in picomolar and nanomolar range for aptamers and CPNs, respectively; amorphous nature of the polymeric matrix; and a likely disordered distribution of aptamers on the nanoparticle surface. Nevertheless, changes in size, changes in the Z potential, changes in the absorption spectra that show the presence of corresponding bands of RNA-aptamers, are a clear indication of the difference between the unconjugated and aptamer-conjugated materials. Moreover, biological assays using not only a tumor cell line for the counterselections of the evaluated aptamers in Cell-SELEX procedure (BT-474), as well as the use of a non-specific aptamer (SCR), further support the successful decoration of the nanoparticles with the specific aptamers.
Although it would be interesting to have a direct observation of the CPN morphology, this information is not critical for the proposed application. Additionally, the conditions required to obtain electron micrographs with sufficient resolution to observe morphological details of nanoparticles having diameters of ~ 20nm are harsh on soft materials such as CP and aptamers. Furthermore, the conventional strategy of metalation might also affect the intrinsic morphology of the material.
2. I noticed that blue light (420 nm) was used for irradiation. This greatly hampers the potential use of this material in studies other than in vitro, due to the low penetration of light of such low wavelength through tissues. Limitations of the study should be discussed and possible solution proposed. Other than that, I did not find the information whether the effect of light alone on cells was tested, without the photosensitizer?
We thank the reviewer for bringing up this point. In order to add information in this aspect, the following paragraph was added in the discussion section.
Paragraph added: It is known that the penetration depth of light in biological tissues is highly dependent on its wavelength. Although light sources centered at 420 nm are not expected to have high penetration depths in biological tissue, the in vivo successful application of PDT protocols using light in this range has been previously reported in several articles [46–48]. The effective penetration of sufficient photons to elicit a reasonable phototherapeutic effect it is a complex combination of absorption, scattering, and diffraction of the target tissue and of the effective fluence of the light source. Furthermore, we envision that CPN functionalized with aptamers developed in this work could be used in PDT protocols as an adjuvant post-surgical treatment (in tissue regions readily accessible for superficial irradiation) to eliminate remaining tumor cells. Taking these observations into account, the limited penetration of blue light into tissue is not expected be an inconvenience.
Regarding the effect of light alone on cells, the results were presented in the supplementary information file accompanying the first version of the manuscript. Please refer to Figure S4.
3. You tested the mediation of singlet oxygen generation, but did you consider that other ROS may be produced, such as for example hydroxyl radical?
We thank the reviewer for his/her suggestion. To address this point, we added the following sentences in the result section.
Paragraph added: Although this work was not aimed at establishing all the possible ROS produced by the particles, we decided to study the formation of 1O2, a highly relevant reactive oxygen species. It is important to note that other ROS species might be forming and playing a role in the observed PDT effect.
4. Typos/English/style: “in vitro ein vivo” (line 88); “2’Fluoro-pyrimidines” – should be 2’-fluoropyrimidines (129); “O-acylisourea” – O should be italicized (156); “4’,6-Diamidino-2- phenylindole” - 4’,6-diamidino-2-phenylindole (238); “citotoxicity” (482); commas are given instead of points as the decimal separator (subchapter 3.5).
We appreciate the reviewer's corrections. We checked for typo/English/style errors and amended them following the track changes function in the revised version of the manuscript.
Round 2
Reviewer 1 Report
The authors have addressed most of my concerns, and I have no more questions.